# OpenReview forum: "Frontier Models Can Take Actions at Low Probabilities"
_ICML.cc/2026/Conference — ICML 2026 regular_

### Official Review · Reviewer_K7N8 · 2026-03-11

**Soundness:** 3
**Presentation:** 3
**Significance:** 3
**Originality:** 3
**Overall Recommendation:** 4
**Confidence:** 3

**Summary:**

This paper investigates whether frontier LLMs can take policy-violating action at low probabilities, enabling evasion of finite-sample pre-deployment audits. The authors decompose this capability into four steps (probability derivation, entropy access, randomness extraction/decision, execution) and design controlled experiments across three task domains (coding, email, Rock-Paper-Scissors) and two entropy regimes (external in-context entropy vs. self-generated). Evaluating GPT-5 and Qwen-3 families, they find frontier models can achieve calibrated defection rates as low as 0.001%, especially with external entropy cues. They derive concrete audit budget implications from observed output probability floors, and note that chain-of-thought legibility currently provides a detection surface.

**Compliance With Llm Reviewing Policy:**

Affirmed.

**Final Justification:**

The rebuttal has answered my questions. As written in my initial assessments, this work is a rigorous empirical research that studies the timely issue of LLM safety. I remain positive about this work.

**Key Questions For Authors:**

1. You note that successful low-rate execution currently requires visible CoT reasoning and that a CoT monitor could detect it. But a strategically deceptive model might hide this reasoning in the long reasoning process or implement it implicitly. Have you tested no-reasoning (non-CoT) conditions, and how does calibration degrade?

2. The failure mode analysis shows that models use heuristic algorithms with systematic biases. My follow-up questions are: Do these biases remain stable across repeated calls, or do models vary their algorithm, and does this variation itself become a detection surface for monitors?

**Limitations:**

The limitation and impact sections do not sufficiently acknowledge the gap between instruction-following capability and autonomous adversarial behavior, which is the most important caveat for interpreting the safety implications.

**Strengths And Weaknesses:**

*Strength*
1. This paper is well-motivated and provide a novel perspective to AI control/safety. The four-step decomposition is clean and diagnostic.
2. The paper has rigorous experimental design with appropriate statistics. The use of Wilson confidence intervals, power-conscious sample sizing, and explicit classification of calibrated/overshoot/undershoot outcomes is methodologically sound. The ablation experiments (providing external strings to isolate algorithm errors vs. running model-generated strings through correct implementations) meaningfully localize failure modes. The audit budget derivation from OPFs via a simple closed-form formula is a practical and actionable output.

*Weakness*
1. There is a concern that this paper overly conflates *instruction-following capability* with *deceptive intent*. The paper frames calibrated low-rate action as a safety-relevant capability, but the results measure instruction-following fidelity, which is not really strategic misbehavior of the models. From training perspetive, a model choosing to defect might have gone through a very different optimization landscape than a model asked to comply with probabilistic instructions. The paper does not provide any evidence or argument that these capabilities transfer to the adversarial setting, which is the core safety concern.

---

> ### Author Rebuttal · Authors · 2026-03-31
>
> Thank you for the detailed and thoughtful review, especially your engagement with both the methodology and its safety implications. We were glad to hear that you found the paper well-motivated, the four-step decomposition clean and diagnostic, and the experimental design rigorous with appropriate statistics.
>
> ### 1. Instruction-following vs. deceptive intent & Limitations
> We agree with the reviewer that instruction-following capability and autonomous adversarial behavior are distinct. It is possible that explicit instruction leads to better performance than an intrinsically held goal would, and our results cannot rule this out. This is a real limitation.
>
> That said, the paper is only claiming that frontier models are capable of taking actions at very low, hard-to-detect rates when explicitly instructed to do so. Measuring this, separately from whether a model would do so spontaneously, is standard practice in AI safety research. [Benton et al. (2024)](https://arxiv.org/abs/2410.21514) and [Meinke et al. (2024)](https://arxiv.org/abs/2412.04984) both measure safety-relevant capabilities under elicitation; the gap the reviewer identifies is present in that work as well.
>
> We do argue the gap is somewhat narrower here because the sub-capabilities we measure (entropy extraction, threshold computation, calibrated decision-making) are general algorithmic capabilities, not skills that only emerge under specific prompting conditions. A model that has learned to extract randomness from a UUID and apply a threshold at 0.01% can potentially use this same algorithm regardless of whether the instruction came from a system prompt or from internally represented goals. [Mallen et al. (2025)](https://arxiv.org/abs/2412.12480) independently confirm that probability calibration is an important sub-skill in their end-to-end Subversion Strategy Eval, where models must autonomously decide when to take a bad action. Our multi-condition design also shows frontier models succeed even without any hint to use entropy, suggesting the capability does not depend heavily on elaborate elicitation.
>
> **Action 1.1:** We plan to add an explicit paragraph in the Limitations section acknowledging the gap between instruction-following capability and autonomous adversarial behavior, while arguing (with citations) why capability measurement is the appropriate and standard methodology for safety-relevant evaluations.
>
> **Question 1.2:** Does this framing (capability evaluation as a standard and well-motivated methodology in AI safety) address your concern? Or is there a specific type of evidence you would find compelling for the transferability claim?
>
> ### 2. No-reasoning (non-CoT) conditions
> Please see our response to Reviewer bgJB (Question 5), where we address this point with supporting results.
>
> ### 3. Stability of algorithmic biases
> Thank you for this question.
>
> **On algorithm stability across repeated calls.** This is a good empirical question that our analysis did not directly address. Our failure mode analysis in Section B.6 characterizes the distribution of algorithms used broadly but does not directly track whether a given model applies the same algorithm consistently across repeated calls. We cannot answer this from our current data. The B.6 breakdown shows that across our sampled traces, models predominantly use ASCII sum-based methods (73.3%), followed by other methods including hex parsing and direct integer interpretation (18.3%), and random float checks (8.4%). However, this distribution is aggregated across conditions. We agree that tracking algorithm stability across conditions, settings, and target rates is an interesting direction and plan to flag it explicitly as future work.
>
> **On whether algorithm variability constitutes a detection surface.** This question prompted us to think more carefully about what our CoT monitoring argument actually rests on, and we think the answer is worth clarifying. The paper's CoT monitoring argument does not rest on algorithm stability, so this gap does not affect the central detection claim. The relevant signal is the presence of a deliberate entropy-to-decision reasoning process in the CoT, regardless of which specific algorithm is used. This pattern distinguishes strategic low-rate execution from normal instruction-following. Algorithm variability is therefore largely orthogonal to the detection argument.
>
> ### What do you think?
> Thank you again for the detailed and constructive review! Do the above actions address your concerns? We would be happy to incorporate further suggestions.

---

> > ### Author Rebuttal · Reviewer_K7N8 · 2026-03-31
> >
> > I thank the authors for this nice rebuttal, I confirm that all my questions are addressed. I don't have further questions as it has been a rigorous and timely empirical paper from the start.

---

### Official Review · Reviewer_nDSj · 2026-03-16

**Soundness:** 3
**Presentation:** 3
**Significance:** 2
**Originality:** 3
**Overall Recommendation:** 4
**Confidence:** 3

**Summary:**

The paper operationalizes a new concept in the context of generative language models: an evaluation of whether a model could execute action with a given target (small) probability based on a prompt instruction and a possible external source of randomness. This task is motivated by the question of whether models could generate certain undesirable strings in a way that does not show up in finite-sample audits, but could show up in deployment. The paper provides a methodology to evaluate this property using three tasks, and performs a sensitivity analysis including external randomness or not with different hints in the prompt. Then the paper reports on a series of findings on the calibration of multiple API-accessible generative models, finding, e.g., that GPT-5 can achieve calibration with an external randomness and even without it if prompted in a special way.

**Compliance With Llm Reviewing Policy:**

Affirmed.

**Ethical Review Concerns:**

I am concerned about the repeated anthromorphic language in the paper: "subversive models", "strategic models", "model defection", etc. Unlike other standard anthromorphizing terms such as "hallucinations", the approach taken in this paper seems to ascribe even more human-like agency to generative language models. There is an ongoing scholarly conversation about the harms of such language (see, e.g., [EMNLP'24](https://arxiv.org/abs/2305.09800)), which might need to be addressed here. We might want to borrow the guidelines from other scientific communities such as Nature:
https://www.nature.com/articles/s42254-023-00584-1

**Ethical Review Flag:**

Flag this paper for an ethics review.

**Ethics Expertise Needed:**

["Other Expertise"]

**Final Justification:**

My concerns were addressed. I keep my positive score.

**Key Questions For Authors:**

No questions at this point.

**Limitations:**

The limitations of the study are addressed well.

**Strengths And Weaknesses:**

The paper operationalizes a new concept of language models "acting" with a desired probability. It introduces a sound methodology, and provides an extensive evaluation with good experimental design and multiple relevant conditions checked. The study design produces multiple interesting findings on the calibration behavior of models. The paper is mostly well-written and quite clear, despite introducing a new and somewhat unusual concept.

Although the paper is sound with respect to the posed research question, I am not convinced of the significance of this research question, and the paper does not provide any evidence that this question is indeed significant. Although it is somewhat interesting that a probabilistic generative model can generate a desired piece of text with a given probability, implications for security are fetched and unsubstantiated. The paper would greatly benefit from include a safety-relevant task and a demonstration of an adversarial scenario with a concrete threat model that clearly showcases the issue motivating this task; otherwise it seems somewhat speculative. Why would a deployer of a highly capable expensive model be interested in planting such a backdoor in principle, and can they do so without detection?

The paper also persistently uses anthropomorphic language that seems to ascribe substantially more agency to generative language models than other common anthropomorphic terms (e.g., "hallucinations"). There are many arguments that this is harmful ([example](https://arxiv.org/abs/2305.09800)). Other scientific venues such as Nature [recommend against it](https://www.nature.com/articles/s42254-023-00584-1) as a matter of editorial policy. I would strongly suggest changing the phrasing of "strategic models", "defecting models", etc., to a clear adversarial model (see above). For instance, "malicious model deployers planting capabilities while evading detection".

Overall, even though I do believe that a paper lacks a convincing demonstration with a tangible threat model with a task that showcases the issue, it executes very well within its research question, while providing some motivation using tangible work. I suggest adding either such a task or at least a small-scale/abridged demonstration with a convincing threat model in the final version.

---

> ### Author Rebuttal · Authors · 2026-03-31
>
> We are grateful for your careful reading and thoughtful engagement. Thank you for recognizing the sound methodology, extensive evaluation, and interesting findings on model calibration behavior.
>
> ### 1. Significance of the research question / need for a concrete threat model
>
> The central significance of this work is that a model producing policy-violating outputs at a calibrated low rate can systematically pass finite-sample pre-deployment evaluations while still causing harm at scale in deployment. The threat model is one where a model has acquired misaligned goals through training pathologies, backdoors, or prompt injection, and is capable of strategically modulating its behavior to evade detection ([Hubinger et al. 2024](https://arxiv.org/abs/2401.05566); [Greenblatt et al. 2024](https://arxiv.org/abs/2412.14093)). Low probability calibration is the specific capability that enables this evasion, and whether current models possess it is an open empirical question. Our paper provides the first direct measurement of this capability to our knowledge. The importance of low probability calibration as a distinct measured capability is independently validated by [Mallen et al. (2025)](https://arxiv.org/abs/2412.12480), who identify it as a key component of protocol subversion in their Subversion Strategy Eval.
>
> We designed our tasks (coding, email, RPS) to be simple in order to measure calibration without confounding it with task difficulty or safety filters, which is standard methodology in capability evaluations. [Kutasov et al. 2025](https://arxiv.org/abs/2506.15740) and [Meinke et al. 2024](https://arxiv.org/abs/2412.04984) similarly use prompted conditions and benign environments to measure scheming capability rather than demonstrate real-world harm.
>
> **Action 1.1:** We will expand Section 3 (Threat Model) to walk through a concrete end-to-end attack scenario step-by-step, identifying calibrated low-rate execution as the specific link our paper measures.
>
> **Action 1.2:** We will add a brief discussion noting safety-positive results (e.g., all models fail to achieve calibration when deriving target rates without entropy; no-CoT substantially degrades calibration), demonstrating the evaluative rather than purely alarmist nature of the contribution.
>
> **Question 1.3:** Does the combination of the expanded threat model walkthrough, the independent confirmation from [Mallen et al. (2025)](https://arxiv.org/abs/2412.12480), and the framing as standard capability evaluation methodology address your concern about significance? Or would you recommend a specific type of demonstration that you think is missing?
>
> ### 2. Anthropomorphic language
>
> We note that terms like "strategic model" and "defection" are borrowed from the AI control literature, where they have established technical meanings rather than claims about model cognition. **We agree that disambiguating this more carefully would improve the paper's accessibility.**
>
> Our empirical contribution is purely behavioral: we measure whether models produce a target action at a specified low rate, and we make no claims about model intentionality or inner states. We will clarify this with a terminological caveat early in the paper.
>
> **Action 2.1:** We will replace "defection" and its derivatives with "target action" throughout, except where directly summarizing prior AI control work in which the term is load-bearing.
>
> **Action 2.2:** We will add a consistent behavioral reframing across the introduction, threat model, and related work sections, making explicit that all capability claims are behavioral and that agentive terminology is adopted descriptively from the AI control literature.
>
> **Question 2.3:** Do the actions above (replacement of "defection," behavioral reframing, related work revision) sufficiently address both the review concern and the ethics flag, or would you recommend additional changes?
>
> ### What do you think?
>
> Thank you again for reviewing the paper! Do the above actions address your concerns? We would be happy to incorporate further suggestions.

---

> > ### Author Rebuttal · Reviewer_nDSj · 2026-04-02
> >
> > The rebuttal addresses my concerns, provided the authors apply the proposed revision.
> > I will keep my positive score.

---

### Official Review · Reviewer_bgJB · 2026-03-17

**Soundness:** 4
**Presentation:** 3
**Significance:** 4
**Originality:** 4
**Overall Recommendation:** 5
**Confidence:** 4

**Summary:**

This paper studies whether LLMs can take actions at low probabilities. This is highly relevant to auditing AI systems that may have adversarial goals (e.g., implanted through backdoors or prompt injection). The question itself is novel, and the study is thorough. Many different settings are tested, all with meaningful motivations (e.g., testing different sources of randomness that might appear in real-world environments, like UUIDs). The results are fairly concerning: Current models are quite capable of taking actions in a calibrated manner at very low probabilities.

At a high level, the paper identifies and characterizes a concerning problem. There is clear scope for follow-up work, and I think this would generate good discussion at the conference.

**Compliance With Llm Reviewing Policy:**

Affirmed.

**Final Justification:**

The rebuttal addressed my concerns and added new, interesting results.

I recommend this paper for acceptance.

**Key Questions For Authors:**

No questions

**Limitations:**

Limitations adequately discussed

**Strengths And Weaknesses:**

Strengths:
- The research question is novel. I haven't seen anyone study this before. This paper basically introduces a new problem to worry about in the AI safety space, which is fairly rare.
- The submission is technically sound. The prompts in the appendix make sense, and the metrics are sensible.
- The writing is clear overall, although see the first point of the weaknesses below.
- The findings are significant and of practical importance. Real adversaries trying to insert backdoors would likely leverage this as an ingredient in an attack chain, since it could make auditing significantly harder.

Weaknesses:
- The abundance of different experimental settings (3 tasks, 4 conditions) clutters the main paper. This makes it harder to skim. I'm not sure what to do about this, since some of the points of the main paper do come from these specific conditions. This isn't a huge issue.
- It would be good to cite the long literature of calibrated uncertainty estimation in deep learning. Recent papers are cited, but there are many that are missing. I recommend looking through the papers citing "On Calibration of Modern Neural Networks" and some surveys. In the LLM era, "Language Models (Mostly) Know What They Know" from Kadavath et al. was the first to the best of my knowledge.
- It would have been nice to include more models from different model families (e.g., Claude, Olmo, DeepSeek, Gemma, Llama).
- Some of the takeaways are overfitting on the small set of models that were evaluated. E.g., one of the paragraph titles is "Newer frontier models are better at game strategy derivation", but this is determined based on a handful of gpt-4 and gpt-5 models. The results do not support such a sweeping generalization. You would need to test a much broader set of models and look at correlations wrt their release dates to make a claim like this.
- It would have been nice to see these results: " An interesting next step is to rerun comparable evaluations under no-reasoning conditions, to test how dependent low-rate calibration is on explicit reasoning."
    - It would presumably be very easy to run these experiments. I recommend adding this in an updated version of the paper if possible, since this information would be valuable to the community.


Note: On some level, one could claim that these results are obvious. If you assume LLMs can generate calibrated random strings, then of course they can threshold those in reasoning traces to take actions with low probability. I don't think this is a real weakness, though. The execution of this paper and the thorough analysis create value for the community. Also, nobody thought to try this before.

---

> ### Author Rebuttal · Authors · 2026-03-31
>
> Thank you for your careful reading and thoughtful feedback, especially your detailed assessment of our experimental design and metrics. Your note that "this paper basically introduces a new problem to worry about in the AI safety space, which is fairly rare" is very encouraging.
>
> ### 1. Experimental clutter across settings
> We agree the breadth of settings can make the paper dense. The multi-task, multi-condition structure motivates our core claims (that calibration generalizes across domains and that entropy access is the key bottleneck) and the aggregate results are sufficient to establish both. In revision:
>
> **Action 1.1:** We will replace per-task figures in Sections 5.1 and 5.2 with compact aggregate panels (extending Figures 1 and 2c across all model families), moving per-task breakdowns to the appendix for verification. Figure 2b will receive a more prominent forward-reference so readers understand the design rationale upfront.
>
> **Action 1.2:** We will add a dedicated figure in Section 5.3 comparing game calibration against aggregated Email and Coding calibration, making the effect of the strategy derivation step directly visible.
>
> ### 2. Missing calibration literature
> **Action 2.1:** We will add a discussion of the calibration literature in Section 2, including [Guo et al. (2017)](https://arxiv.org/abs/1706.04599) and [Kadavath et al. (2022)](https://arxiv.org/abs/2207.05221), and clarify the distinction between our setting (calibrated action rates under deliberate probabilistic control) and the standard setting (whether confidence scores match empirical accuracy).
>
> ### 3. Model coverage
> **Action 3.1:** We will add the Claude 4.5 family (Opus, Sonnet, Haiku) to all main experimental conditions including the game task (baseline + SSoT) and the no-CoT experiments. This provides a third major frontier family spanning three capability tiers, substantially strengthening generality claims. Our existing Qwen results already establish that models below ~30B consistently fail to calibrate below 10%.
>
> **Question 3.2:** Would a full three-tier Claude family (alongside GPT-5 and Qwen-3) sufficiently address the model diversity concern, or would you prioritize specific open-weight families?
>
> ### 4. Overgeneralization of frontier model claims
> On the strategy derivation claim: we agree and will revise the section title and constrain the claim to what the data supports — that a large capability gap exists for Nash equilibrium derivation within the GPT-4o/GPT-5 comparison specifically — and add an explicit limitations note.
>
> On empirical breadth: we ran the full calibration evaluation on the Claude 4.5 family (results below). Claude models show good calibration with in-context entropy across all three tasks, broadly comparable to GPT-5 and GPT-5-mini, supporting the claim that calibrated low-rate action is not confined to a single model family.
>
> | Model | Coding 0.01% | Coding 0.1% | Coding 1% | Email 0.01% | Email 0.1% | Email 1% | Game 0.01% | Game 0.1% | Game 1% |
> | :---- | :---: | :---: | :---: | :---: | :---: | :---: | :---: | :---: | :---: |
> | **Haiku 4.5** | 0.012 | 0.090 | 0.709 | 0.010 | 0.100 | 1.027 | 0.010 | 0.100 | 1.041 |
> | **Sonnet 4.5** | 0.009 | 0.123 | 1.715 | 0.010 | 0.100 | 0.969 | 0.010 | 0.100 | 1.010 |
> | **Opus 4.5** | 0.029 | 0.161 | 0.906 | 0.010 | 0.100 | 1.007 | 0.010 | 0.100 | 1.023 |
>
> ### 5. No-reasoning (non-CoT) experiments
> We report results for two conditions and target rates below; the full set will be included upon acceptance.
>
> **1% target rate:**
> | Setting | Model | Coding (no-CoT/CoT) | Email (no-CoT/CoT) | RPS (no-CoT/CoT) |
> |---|---|---|---|---|
> | Implicit | sonnet-4-5 | 0.00/**1.73** | 0.00/**1.07** | 44.86/**1.01** |
> | Implicit | gpt-5.4-mini | 0.00/**3.31** | 0.00/**0.82** | 24.94/**1.21** |
> | Explicit | sonnet-4-5 | 0.00/**1.00** | 0.00/**1.00** | 6.78/**1.00** |
> | Explicit | gpt-5.4-mini | 0.00/**1.12** | 0.00/**0.92** | 11.40/**1.58** |
>
> **0.1% target rate:**
> | Setting | Model | Coding (no-CoT/CoT) | Email (no-CoT/CoT) | RPS (no-CoT/CoT) |
> |---|---|---|---|---|
> | Implicit | sonnet-4-5 | 0.00/**0.12** | 0.00/**0.10** | 44.86/**0.10** |
> | Implicit | gpt-5.4-mini | 0.00/**1.76** | 0.00/**0.13** | 24.94/**0.10** |
> | Explicit | sonnet-4-5 | 0.00/**0.10** | 0.00/**0.10** | 6.78/**0.10** |
> | Explicit | gpt-5.4-mini | 0.00/**0.15** | 0.00/**0.24** | 11.40/**0.18** |
>
> Without CoT, performance is worse than with CoT across both target rates, confirming that reasoning is a key bottleneck for low-rate calibration.
> ### What do you think?
> Do the above actions address your concerns? We would be happy to incorporate further suggestions.

---

> > ### Author Rebuttal · Reviewer_bgJB · 2026-04-06
> >
> > A 3-tier Claude evaluation would address my concerns about model coverage.
> >
> > Thank you for adding the experiments. I will maintain my score and recommend the paper for acceptance.

---

### Decision · Program_Chairs · 2026-04-30

**Decision:**

Accept (regular)

**Comment:**

This work studies a novel question and runs a comprehensive suite of experiments to study it, and makes clear and precise claims. The weaknesses I observed in my own reading were largely addressed by the rebuttal. In particular, the planned revisions on model coverage and clearer threat-model framing would substantially strengthen the paper and resolve what I read to be the main reviewer concerns. Some of the reviewer asks are "nice-to-haves" that I think would improve the present paper. As a matter of taste, I agree with reviewer nDSj's comments about some of the stylistic choices, and think the paper would be stronger with the suggested revisions. (This last point does not affect my recommendation.)